

# Design and Evaluation of a Thermal Precipitation Aerosol Electrometer (TPAE)

Shipeng Kang[1,2], Tongzhu Yu[1], Yixin Yang[1], Jiguang Wang[5,6], Huaqiao Gui[1,2,4], Jianguo Liu[1,2,4], Da-Ren Chen[3]

[1]Key Laboratory of Environmental Optics and Technology, Anhui Institute of Optics and Fine Mechanics, Hefei Institutes of Physical Science, Chinese Academy of Sciences, Hefei 230031, China

[2]University of Science and Technology of China, Hefei 230026, China

[3]Particle Laboratory, Department of Mechanical and Nuclear Engineering, Virginia Commonwealth University, Richmond, VA 23284, USA

[4]Innovation excellence Center for Urban Atmospheric Environment, Institute of Urban Environment, Chinese Academy of Science, Xiamen 361021, China

[5]China Automotive Technology and Research Center Co., Ltd, No.68 Xianfeng East road, Dongli District, Tianjin, 300300, China

[6]CARARC Automotive Test Center (Kunming) Co., Ltd, No.68 Donghuan road, Kunming, 651701, China

*Correspondence to*: Da-Ren Chen (dchen3@vcu.edu), Tongzhu Yu (tzyu@aiofm.ac.cn)

**Abstract.** Aerosol electrometers are widely applied to measure the number concentration and other physical parameters of particles. In this work, a new aerosol electrometer, i.e., Thermal Precipitation Aerosol Electrometer (TPAE), was designed and its performance was experimentally evaluated. The TPAE integrates a thermal precipitator with a micro- current measurement circuit board for measuring electrical charges carried by particles. The thermal precipitator was in the disk-to-disk configuration. Heating paster and air cooling were adopted to establish a temperature gradient in the precipitator. At a sample flow rate of 0.3 L/min and temperature gradient of 254 K/cm, the precipitation efficiency of particles reaches ~100%. The measurement range of the designed aerosol electrometer was $\pm5\times10^5$ fA, with the accuracy of $\pm2$fA. In the evaluation, the electrical performance of TPAE was evaluated using DMA (differential mobility analyzer)-classified sodium chloride and soot particles and compared it to that of the reference faraday-cage aerosol electrometer. The precipitation performance of TPAE was then studied as the functions of temperature gradient, sampling flowrate and particle size. It is shown that the particle collection efficiency of built-in thermal precipitator is inversely proportional to the sampling flow rate, and proportional to the temperature gradient. The effect of particle size on the above efficiency was minor for sodium chloride particles. Different from that observed for NaCl particles, the slightly positive correlation of the collection efficiency with the electrical mobility size was observed for soot particles (in the size range of 30 ~ 160nm). It might be due to soot agglomerates. The designed



aerosol electrometer with the soft particle collection could be used in, e.g., field studies requiring to both measuring the particle

charges and collecting particles for offline morphology, chemistry, and other studies.

## 1 Introduction

Instruments for measuring the integral parameters of aerosol particles, e.g., total number, surface area, and mass concentration

of particles, is important for the characterization of particulate matters (PM) emitted from various sources. Example

applications of integral parameter instruments are the measurements of particle emissions from vehicles (Faxvog et al., 2007,

Kheirkhah et al., 2020), ocean aerosols (Held et al., 2011), atmospheric aerosols (Hillemann et al., 2014) and urban particles

(Molgaard et al., 2013, Etzion et al., 2018, Alas et al., 2019). More, they could integrate with size classifiers, e.g., differential

mobility classifiers, for characterizing the size distribution of aerosol particles. An example of the integration is the

development of electrical mobility particles sizers, e.g., scanning mobility particle sizers, which are widely applied for

measuring the size distribution of fine and ultrafine particles.

For the characterization of total number concentration of aerosol particles, both condensation particle counters (CPCs) and

aerosol electrometers (AEs) are typically applied. CPCs count the number of particles per a given time by enlarging the particle

size (via the condensation of working fluid vapor) and one-by-one counting them (via an optical means). The single particle

counting of CPCs make them basic instruments for measuring the number concentration of particles. CPCs are often selected

to work with DMAs (differential mobility analyzers), e.g., scanning mobility particle sizers (SMPSs), for measuring the size

distribution of particles. Alternatively, the above measurement task can be accomplished by electrical means, which requires

particles to be electrically charged and collected prior to the electrical charge/current measurement. This type of aerosol

instruments is called an aerosol electrometer (AE). AEs typically work with aerosol chargers to measure the particle number

concentration. Another important usage of AEs is to calibrate the counting efficiency of CPCs (Giechaskiel et al., 2009). An

example of this aerosol electrometer is the one used in electrical aerosol analyzers (EAAs, Liu and Pui, 1975). A commercial

version of the EAA aerosol electrometer is the early generation of TSI Model 3068B.

A Faraday cage equipped with high efficiency filters is typically designed in an aerosol electrometer to completely collect

sampled particles in different sizes and to induce the current resulting from the continuous collection of charged particles. Yang

et al. (2018) developed an aerosol electrometer in which particles were collected by a metal filter and then the current was

directed to a micro current measurement circuit (through a copper probe). The DiSC developed by Fierz et al. (2018) used two

filter stages to collect particles of different sizes, in which a porous metal filter is used in the first stage to collect small particles.

Liu et al. (2020) developed a mini-eUPS in which a miniature aerosol electrometer was used after a plate electrical mobility

classifier to detect the number concentration of classified particles. A TEOM filter disk is used in the miniature aerosol

electrometer. Seol et al. (2000) developed a Faraday cup electrometer for the operation at the pressure of 200-930 Pa, in which



porous metal mesh and filter are used for collecting charged particles. Intra et al. (2014) applied an aerosol electrometer to measure atmospheric ions and charged particles, in which the particle collection was again realized by a HEPA filter. Charged particles could also be collected by the inertial impaction on electrically isolated metal substrates, from which the current resulted from the continuous collection of charge particles is directly measured. An example of such aerosol instruments is an Electrical Low-Pressure Impactor (ELPI) (Keskinen et al., 1992). More, charged particles can be collected by electrical means.

An example of such aerosol instruments is an Engine Exhaust Particle Sizer (EEPS) presented by Tammet et al. (2002) and Wang et al. (2016). However, the collection of charged particles either by the filtration, inertial impaction, or electrical means makes it possible to alter the morphology of sampled particles, particularly for particle agglomerates, such as soot particles. The above issue may not favor the off-line SEM analysis if required.

     Compared to the above collection methods, the collection of particles by the thermal precipitation is a good candidate for the

"soft" collection of particles, e.g., disk-to-disk (Kethley et al., 1952, Wang et al., 2012), plate-to-plate (Tsai and Lu, 1995) and cylindrical thermal precipitators (Bredl and Grieve, 1951, Wang et al., 2012). Furthermore, the minor particle size effect on the particle collection by the thermal precipitation has been documented (Wang et al., 2012). Note that the collection by the inertial impaction is favored for large particles, and the electrical collection is favored for small particles. Both collection methods significantly depend on the particle size.

In a thermal precipitator, particles are introduced into a precipitation zone in which a temperature gradient is established. The direction of temperature gradient is typically perpendicular to that of flow direction. Once entering the precipitation zone, the thermophoretic force moves particles in the direction of temperature gradient, and eventually precipitates them. By the thermal means, particles are collected on the cold plate/electrode. If the cold electrode could be electrically isolated from other metal parts and well insulated, it could serve as an electrode for the current measurement, which favors the cold environment to

reduce the thermal noise. Unfortunately, precipitation plates of existing thermal precipitators could not be directly connected to electrometers due to their poor electrical insulation. The design and cooling method of thermal precipitators must be modified to enable them softly collecting charged particles while measuring the particle current.

     The objective of this work is thus on the development of a thermal precipitation aerosol electrometer (TPAE), integrating a thermal precipitation with the direct current measurement in one device. The overall performance of the protype was

experimentally calibrated and compared with it given by a Faraday-cage aerosol electrometer. For the electrical performance evaluation, the zero- point and response time of the electrometer was calibrated, and the linear correlation of readouts of TPAE and the reference was examined. For the thermal precipitation performance, the collection efficiency of TPAE was investigated as the functions of temperature gradient, sampling flow rate and particle sizes. In addition to NaCl particles, soot particles were also used as test particles.





## 2 Design of Thermal Deposition Aerosol Electrometer


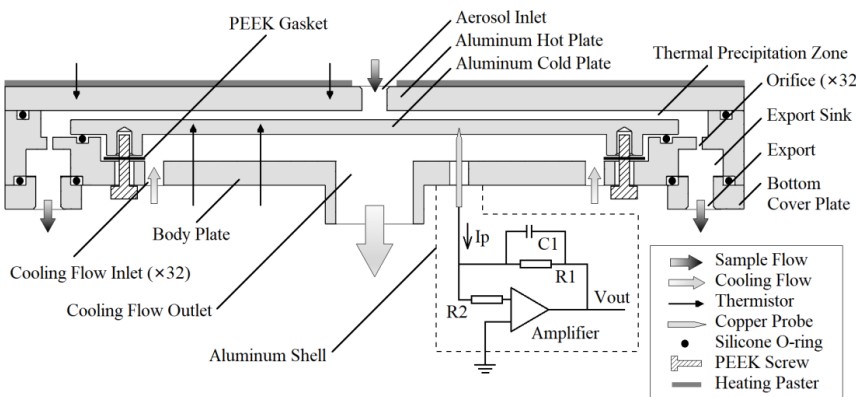

**Fig 1: Schematic diagram of the protype Thermal Precipitation Aerosol Electrometer (TPAE)**

Fig. 1 show the schematic diagram of the thermal precipitation aerosol electrometer (TPAE). It consists of two parts: one part is for the thermal deposition of particles and the other is for the measurement of current carried by continuously collected

particles. The thermal precipitation part is in the disk-to-disk configuration. The sampled flow entered the precipitation part from the inlet tube located at the disk center, radially flew outwards in the space defined by two separated disks aligned at their centers, and eventually exited from the outlet (i.e., a series of holes evenly distributed in the circumferential chamber designed at the disk edge). A temperature gradient was established between two disks (with the top one heated and the bottom one cooled). The spacing between two disks was controlled by PEEK gaskets. To improve the precipitation efficiency of particles

and reduce the size of TPAE, the constructed thermal precipitation zone is 120 and 0.5 mm in height and diameter, respectively. In the thermal precipitation zone, sampled particles were deflected from the flow direction due to the temperature gradient. The top disk was heated by attaching a heating paster on its outside while the cold disk was cooled by air flowing in the adjacent chamber. Driven by the suction from the outlet of the cooling air chamber, the cooling air entered the chamber from a series of holes located close to the chamber outer diameter. The flow rate of cooling air (~ 20 lpm) was monitored by a mass

flow meter (Beijing Sevenstar Flow, Model CS100). Four thermistors (Songtian Electronics, 100KΩ) were also installed to measure the temperatures of heat and cold disks.

A solid copper pin was attached to the cold disk for measuring the current carried by continuously collected particles. Under the above arrangement, the cold disk is served as an electrode, which is enclosed in the cage formed by the hot metal plate and the air-cooling chamber. The cage also encloses the pre-amplifier preventing it from the interference of the ambient

electromagnetic waves. Note that the copper pin, directing the particle current to the micro- current circuit board, is exposed to the cooling air. The thermal noise of the pre-amp could be reduced if cold air was applied. As shown in the dashed line of Fig. 1, the current carried by charged particles was measured through a $R_1$ resistor of 10GΩ so that Vout, the output pin of





the pre-amplifier (ADA4530-1), is by the current. The supply voltage of the pre-amplifier is ±5V, resulting in the measurement range was ±5V/10GΩ = ±5×10$^5$ fA. The capacitor of $C_1$ (47pF) was used to suppress the noise bandwidth. The current and

the number of sampled particles (assuming all the particles carrying the same charges) could be calculated by Eqs. (1) and (2), where $\Delta I_p$ is the increment of discharge current, $\Delta V_{out}$ is the increment of the pre-amplifier output, $\Delta N_p$ is the increment of the number of collected particles, $e$ is elementary charge, $x$ is the average charge of charged particles, and $\eta$ is the particle collection efficiency.

$$\Delta I_p = -\frac{\Delta V_{out}}{R_1} \ , \tag{1}$$

$$\Delta N_p = \frac{\int \Delta I_p dt}{ex\eta} \ , \tag{2}$$

In practical terms,

$$\begin{cases} \Delta I_p = I_{pm} - I_0 \\ \Delta V_{out} = V_{outm} - V_0 \\ \Delta N_p = N_{pm} - N_0 \end{cases} , \tag{3}$$

where $I_{pm}$, $V_{outm}$ are measured values while $I_0$, $V_0$ and $N_0$ are the null point value of thermal precipitation aerosol electrometer. The zero-point, $x$, $\eta$ are unknown parameters to be determined by experiments. To calculate $N_{pm}$ in the

measurements, the zero-point must be measured under the free particle flow condition (i.e., with a HEPA filter placed at the inlet of TPAE), $x$ is determined by the charging of sampled particles, and $\eta$ shall be experimentally calibrated using test particles.

## 3 Experimental Setup and Data Analysis

To evaluate the performance of the TPAE, the basic performance of electrometer and the particle collection efficiency of

thermal precipitation zone should be first investigated. The basic performance of the electrometer includes the zero-point, low response time and its output linearity with the readout of reference electrometer.

### 3.1 Experiment Setup

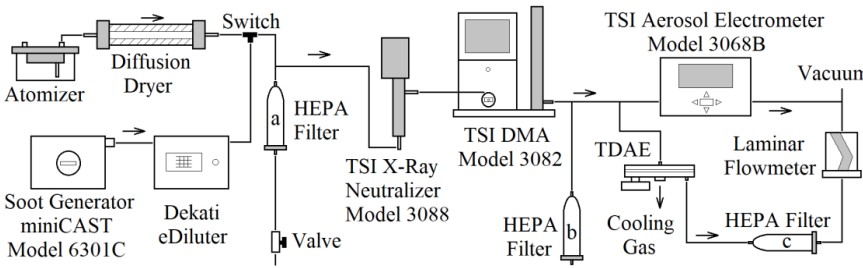





**Fig 2: Schematic diagram of experimental setup**

Fig. 2 shows the experimental setup for investigating the performance of TPAE. The aerosol electrometer, AE, (TSI Model 3068B) was selected as the reference in the setup. NaCl droplets were generated by the atomizer (TSI Model 9302) with aqueous NaCl solutions of 0.005g/ml and water in droplets was removed in a diffusion-type dryer. Soot particles were generated by a soot generator (Jing, Model miniCAST 6301C) with propane as the fuel. A diluter was applied to reduce the concentration of soot particles. A differential mobility analyzer (DMA, TSI Model 3082) was utilized to classify particles of

selected electrical mobility sizes. Particles entering the DMA were passed through a soft X-ray aerosol charger (TSI Model 3088). A bypass line with a HEPA filter and a valve was included in the setup prior to the charger to make sure desired flow rate entering the charger and DMA. The prototype and a Faraday-cage aerosol electrometer (TSI Model 3068B) were connected in parallel at the downstream of the DMA. The other bypass line with a HEPA filter was also included to ensure the total flowrate required for the prototype and reference operation. The sampled particle-laden flow was driven by a vacuum pump

and monitored by a laminar flowmeter. The sampled flow after the TPAE was passed through a HEPA filter to remove uncollected particles prior to the laminar flowmeter. The cooling air of TPAE was driven by the suction from the outlet of the cooling air chamber via a pump. A separate vacuum pump was used to drive the flow for the reference AE. Note that the flow rates of TPAE and TSI electrometer were kept the same in experiments. The $V_{outm}$ of TPAE was converted to digital signals which is sent to a PC so that $I_{pm}$ could be recorded in real time at 4Hz and averaged to 1Hz.

**3.2 Experimental design**

**3.2.1 Measurement of Zero-point**

A stable zero-point is the premise for the measurement by any electrometer. The high fluctuation of zero-point increases the threshold signal-to-noise ratio, making the measurement less sensitive, and the drifting of the zero-point results in inaccurate measurements. An experiment using the above-described setup was carried out to measure the zero-point trend during the

electrometer warm-up. In this part of experiments, air flow was completely drawn from the bypass line installed after the DMA by turning off the soot generator and closing the valve in the bypass line installed before the soft X-ray aerosol charger. The sampling flow rates of TPAE was set at 0.3 L/min. The cooling air flow rate was at ~20 L/min. According to our preliminary experiments, particles in sizes smaller than 100 nm were totally collected under the above settings. It was thus selected as a typical working condition of TPAE. Note that, the zero-point could be varied by the ambient temperature and other parameters.

Zero points should be measured prior to each run.

**3.2.2 Experiment of Response Time**

A step response experiment was performed to measure the response time of TPAE. A fast response time (i.e., less than 1s) is





necessary for high time-resolution measurements, for example, the time variation of number concentration of engine exhaust particles. For an aerosol electrometer, the response time are affected by the rate of particle collection and the characteristics of

micro current measurement circuit. In this part of the calibration, the soot generator was used. The temperature gradient of TPAE was set at 254 K/cm. The sample flowrate of TPAE was the same as that of reference AE. The response time of TPAE was experimented under two working conditions, 0.3 L/min to achieve the full precipitation efficiency (~ 100%) and 0.6 L/min to study the effect of sampling flow rate on the TPAE's collection efficiency. After the warm-up of two aerosol electrometers, a switch valve (between the atomizer and soot generator lines) was used to manually realize the step change in number

concentration of test particles. The DMA was set to classify soot particles with electrical mobility size of 70 nm.

### 3.2.3 Investigation of the linearity of two aerosol electrometer readouts

This part of experiments calibrated the linearity of both TPAE and reference AE readouts. The collection efficiency of the HEPA filter used in the reference AE is close to 100%. The particle collection efficiency of TPAE was assumed constant with given temperature gradient and sampling flowrate according to the previous works (Wang et al., 2012). For this calibration,

soot particles in various concentrations were produced. The electrical mobility size of test particles was 70 nm (classified by the DMA). The temperature gradient of TPAE was maintained at 254 K/cm and the sampling flow rates of both TPAE and reference AE were set at 0.3 L/min. For each test concentration, average of readouts in one minute was reported for the comparison.

### 3.2.4 Study of the particle collection Efficiency of TPAE

For this part of study, the collection efficiency of TPAE was measured as the function of sampling flowrate, temperature gradient, and electrical mobility size, i.e., $\eta(Q_{in})$, $\eta(\overrightarrow{\nabla T})$ and $\eta(d_p)$, respectively, where $\overrightarrow{\nabla T}$ is the temperature gradient of thermal precipitation field, $d_p$ is electromigration particle diameter and $Q_{in}$ is the sampling flow rate. For the measurements of $\eta(Q_{in})$ and $\eta(\overrightarrow{\nabla T})$ of the prototype, sodium chloride particles in the electrical mobility diameters ranging from 23 ~ 200nm was tested, while for $\eta(d_p)$, both sodium chloride and soot particles were studied.

By keeping the same length of transport tubes connecting from DMA exit to both TDAE and reference AE, it is assumed that the particle loss in the tubes were the same since the sampling flow rates of both electrometers was kept the same for a given test. The collection efficiency, $\eta$, was then calculated by Eq. (4), where $I_s$ is the current measured by the reference electrometer.

$$\eta = \frac{I_{pm} - I_0}{I_s} ,$$                    (4)

The temperature gradient, $\overrightarrow{\nabla T}$, was calculated by Eq. (5), where $W$ is the height of the thermal precipitation zone, and the direction of $\overrightarrow{\nabla T}$ is perpendicular to disks from the hot one to the cold one.





$$|\overrightarrow{\nabla T}| = \frac{T_{hot} - T_{cold}}{W} \, , \tag{5}$$

Varying the $\overrightarrow{\nabla T}$ could be realized by changing the cooling air flowrate. The electrical mobility diameters, $d_p$, of test particles was determined by the DMA operation, whose ratio of sheath flow to aerosol sampling flowrates was 4:1. The sampling flowrate of TPAE, $Q_{in}$, was controlled by a valve and monitored by a laminar flowmeter. Table 1 summaries the experimental conditions for this part of study.

**Table 1: Experimental conditions for the measurements of TPAE collection efficiency**

| Measured | Experimental Condition | | | |
|---|---|---|---|---|
| | $d_p$ (nm) | $|\overrightarrow{\nabla T}|$ (K/cm) | $Q_{in}$ (L/min) | Material |
| $\eta(Q_{in})$ | 70, 200 | 254 | $0.3 \sim 1.0$ | NaCl |
| $\eta(\overrightarrow{\nabla T})$ | 70 | $160 \sim 310$ | 0.3, 0.6 | NaCl |
| $\eta(d_p)$ | $23 \sim 200$ | 254 | 0.3, 0.6 | NaCl, Soot |

### 3.3 Model for the particle collection efficiency of thermal precipitation

To validate the measured particle collection efficiency of TPAE, we applied the model developed by Wang et al. (2012) to calculate the thermal deposition efficiency of thermal precipitators in disk-to-disk configuration and compare them to the measured. The detail of the model can be found in the work of Wang et al. (2012). A summary of the model was given for the reference. With assumption that the flow is steady-state, incompressible, laminar and axisymmetric, particles are evenly distributed at the entrance, the collection efficiency of thermal precipitators in the disk configuration can be calculated as

$$\eta = \frac{\pi r^2 V_{th}}{Q_{in}} \, , \tag{6}$$

where $Q_{in}$ is the aerosol sampling flowrate, r the radius of the precipitation disk, and the thermal velocity, $V_{th}$, is calculated as

$$\vec{V}_{th} = \frac{\mu \overrightarrow{\nabla T} H C_c}{\rho_g T} \, , \tag{7}$$

in which $\overrightarrow{\nabla T}$ is the temperature gradient, $T$ is the absolute temperature of particles, $\rho_g$ is the density of carry gas, and $H$ is the thermophoretic coefficient. According to Talbot et al. (1980), $H$ can be calculated by Eq. (8), where $k_g$ and $k_p$ are the conductivity of air and particle. $C_s = 1.147$, $C_t = 2.20$ and $C_m = 1.146$ are constants. $C_c$ is the Cunningham correction factor calculated by Eq. (9).

$$H = \frac{2 C_s (\frac{k_g}{k_p} + C_t K_n)}{(1 + 3 C_m K_n)(1 + \frac{2 k_g}{k_p} + 2 C_s K_n)} \, , \tag{8}$$



$$\begin{cases} C_c = 1 + K_n[\alpha + \beta \exp\left(-\frac{\gamma}{K_n}\right)] \\ \qquad K_n = \frac{2\lambda}{d_p} \end{cases},$$ (9)

where $\alpha = 1.142$, $\beta = 0.558$, $\gamma = 0.999$, $K_n$ is Knudsen number and $\lambda$ is the mean free path of air.

According to Eq. (3-3), the collection efficiency, $\eta$, is inversely proportional to and $Q_{in}$.

**4. Results and Discussion**

**4.1 Performance of electrometer**

**4.1.1 Zero-point measurement:**

Fig. 3 shows the readouts of TPAE during the warm-up. For the reference, the temperatures of hot and cold disks were also given in the figure. At the time zero, the zero-point of the electrometer was approximately 57 fA, and the temperatures of hot

and cold disks were both 22.3℃. During the warm-up, temperatures of plates rose at different rates, established the increasing temperature gradient. The disk temperatures eventually stabilized at 61.40 ℃ (±0.20 ℃) and 48.15 ℃ (±0.15 ℃), respectively. In the meantime, the TPAE readout reduced and finally stabilized at -20.68fA (±2fA). The warm-up of the prototype took approximately 40 minutes and eventually established the temperature gradient of 254 K/cm (±7 K/cm). A higher temperature gradient could be realized by increasing the heating and cooling powers.

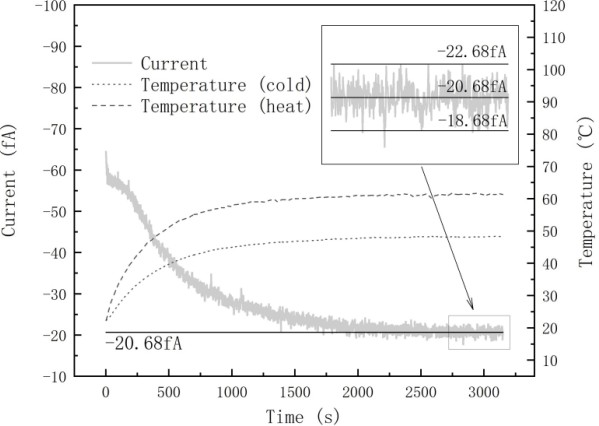


**Fig 3. The readout of TPAE during the warm-up process (particle-free air was used).**

Assuming that particles in sample gas were singly charged, the sampling flowrate is $Q$, and $e$ is elementary charge, the current $I$ can be calculated as $eNQ$, where $N$ is the particle number concentration. If $Q = 0.3$ L/min, $N = 1,250 (\# \cdot cm^{-3})$ per fA. The $\pm 2$fA fluctuation is equivalent of $\pm 2,500$ #/cm³.



**4.1.2 Response time measurement**

Fig. 4 shows the readout of TPAE experiencing a step change in the number concentration of soot particles at the sampling

flowrates of 0.3 L/min (a) and 0.6 L/min (b). For the reference, the readout of reference electrometer is also included in the

same figure. It is found that the trends of TPAE and reference electrometer are consistent. Ideally, $I_{\text{TPAE}}(t)$ equals to $\eta I_{3068}(t)$,

$t > 0$, where $\eta = I_{\text{TPAE}}(t)/I_{3068}(t)$ is the particle collection efficiency of TPAE. The efficiency data as a function of time is

further included in the figure. In the time periods of $0 \sim 30s$ and $140 \sim 170s$ in the case of 0.3 L/min, and the periods of $0 \sim 16$

s and $80 \sim 100$ s in the case with 0.6 L/min, the current readouts were very low (closed to zero), resulting in the unsteady $\eta$.

In the time periods of $40 \sim 120s$ and $25 \sim 70s$ for the sampling flowrates of 0.3 and 0.6 L/min, respectively, the collection

efficiency was constant. It is because the number concentration of test particle was stable and the $\eta$ was kept constant (i.e.,

98.5% (±1%) for 0.3 L/min case, and 56.0% (±1%) for 0.6 L/min case).

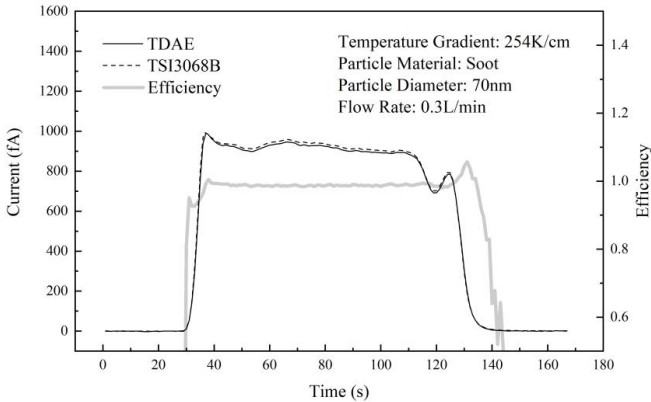


**(a) sampling flow: 0.3 L/min**

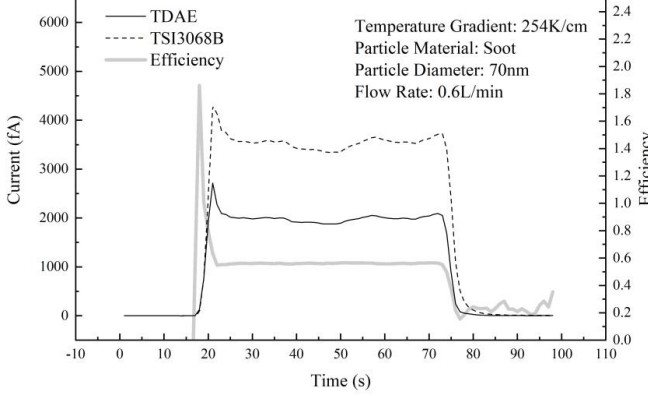

**(b) sampling flow: 0.6 L/min**

**Fig 4: The readout of TPAE in response to the step particle concentration change. The readout of reference AE was also included as**

**the reference.**





For sampling flowrate of 0.3 L/min, the collection efficiency in the period of 30 ~ 40s was less than 98.5%, indicating the response of TPAE to step rise of particle concentration is slower than that offered by reference electrometer. The same observation could be found in the period of 125 ~ 135s. However, the response time difference between two electrometers is within one second. The similar conclusion can be reached in the case with the sampling flowrate of 0.6 L/min. Therefore, the

response of TPAE can keep up with that of reference electrometer within 1s.

### 4.1.3 Readout linearity between two aerosol electrometers

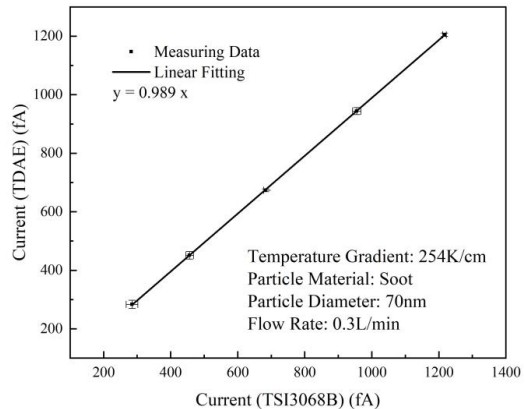

**(a) sample flow: 0.3L/min**

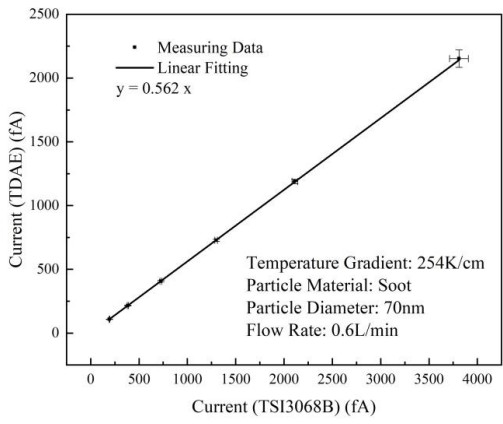

**(b) sample flow: 0.6L/min**

**Fig 5: Linear correlation between the readouts of TDAE and reference AE**

Figure 5 shows the readout correlation between TPAE and reference electrometer at the sampling flow rates of 0.3 and 0.6 L/min. The linear correlation between two readouts was observed. In the case of 0.3 L/min (Fig. 5a), the best linear fitting resulted in the slope of 0.989. Note that the slope of this best linear fitting is the particle collection efficiency of TPAE. It is

because the collection efficiency of reference electrometer is close to 100%. Similarly, in the case of 0.6 L/min (Fig. 5b), the



best fitting with a straight line obtained the slope of 0.562. The above observation of reduced particle collection efficiency

with the increase of sampling flowrate is expected according to Eq. (3-3).

**4.2 Investigation of TPAE particle collection efficiency**

**4.2.1 Effect of temperature gradient**

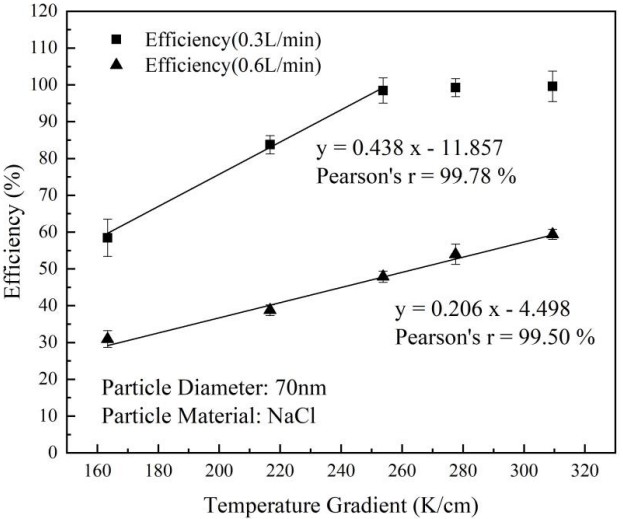


**Fig. 6: Calibration of collection efficiency and temperature gradient**

Fig. 6 shows the particle collection efficiency of TPAE as the function of temperature gradient for NaCl particles of 70 nm in

size and at the sampling flow rates of 0.3 and 0.6 L/min. It is found that, for 0.3 L/min flowrate, the collection efficiency is

linearly increased with the increase of temperature gradient, and the collection efficiency reached ~ 100% at the gradient

exceeded 254 K/cm. For the flowrate of 0.6 L/min, the collection efficiency was again linearly increased with the increase of

temperature gradient within the test range. The above experimental observation is consistent with that given by Eq. (3-3)





**4.2.2 Effect of sampling flow rate and particle size**

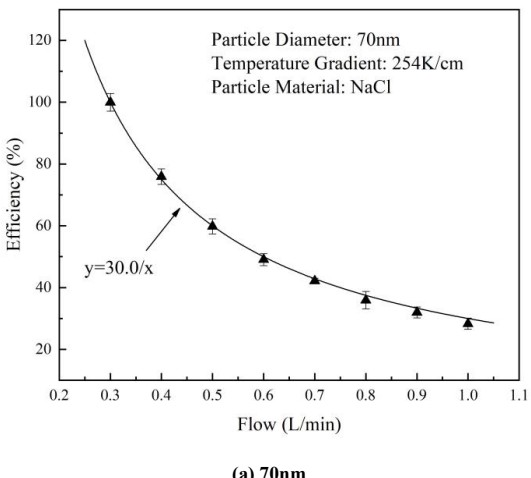

**(a) 70nm**

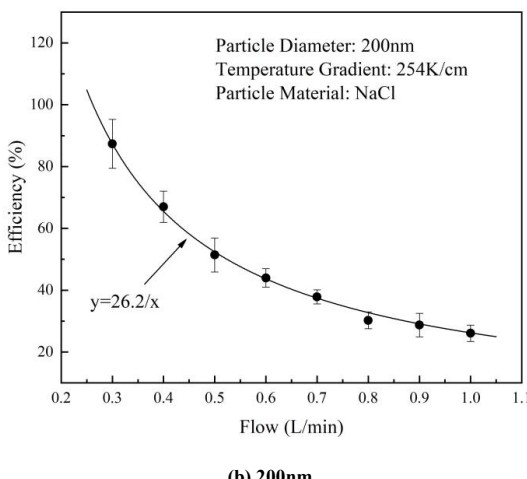


**(b) 200nm**

**Fig 7: The particle collection efficiency of TPAE as the function of sampling flowrate at three different NaCl particles sizes (i.e., 70 and 200 nm)**

In this part of experiments, the temperature gradient in TPAE was set at 254 K/cm and NaCl particles of 70 and 200 nm in

sizes were selected for this investigation. The sampling flowrate of TPAE was varied from 0.3 to 1.0 L/min. Fig. 7 shows the

measured particle collection efficiency of TPAE as the function of sampling flow rate for a given particle size. As expected,

for a given particle size, the collection efficiency was reduced as the increase of sampling flow rate, and the reduction

characteristics follows the Eq. (3-3), i.e., the collection efficiency is inversely proportional to the sampling flow rate ($Q_{in}$). As

a result, the products of $(\eta \cdot Q_{in})_{70nm}$ and $(\eta \cdot Q_{in})_{200nm}$ remained $30.0 \pm 1.7\% \cdot L/min$, and $26.2 \pm 2.0\% \cdot L/min$,

respectively.





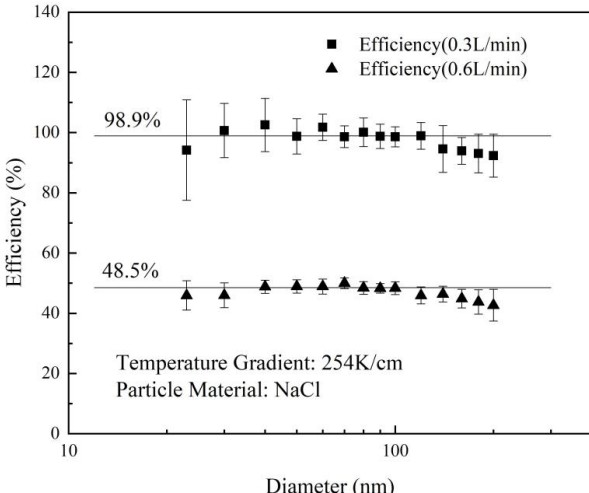

**Fig 8: The measured TPAE particle collection efficiency as a function of the NaCl particle size for the sampling flow rate of 0.3 and 0.6 L/min.**

The effect of particle size on the TPAE collection efficiency is given in Fig. 8 for the sampling flow rates of 0.3 and 0.6 L/min. For the particle size less than 120 nm, the collection efficiency was ~ 98.9% and ~ 48.5% for both 0.3 and 0.6 L/min flow rates, respectively. As the particle diameter further increased, the collection efficiency of TPAE was slightly decreased, which is consistent with the thermal precipitation velocity obtained in previous works (Beresnev et al., 2019, Wang et al., 2012). According to Eq. (3-4), the larger the particle size the lower the thermal precipitation velocity

**4.2.3 Collection efficiency for soot particles**

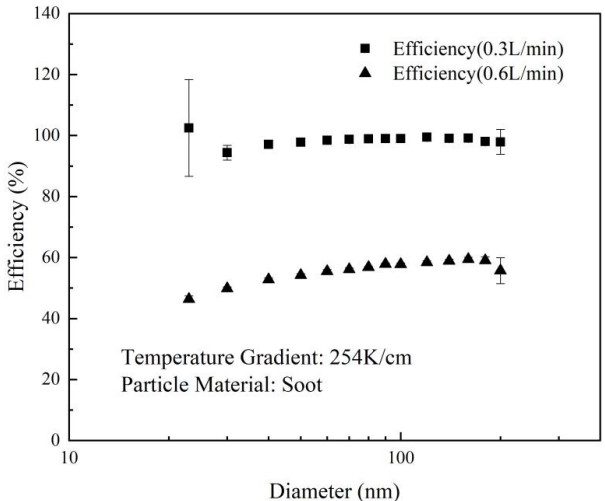





**Fig 9: The measured TPAE particle collection efficiency as a function of the soot particle size for the sampling flow rate of 0.3 and 0.6 L/min.**

In addition to NaCl particles, soot particles were also selected in the collection efficiency measurement. The electrical mobility
size of soot particles in the range of 23-200 nm was tested. The measured particle collection efficiency as a function of electrical

mobility size at the temperature gradient of 254 K/cm and the sampling flow rates of 0.3 and 0.6 L/min is given in Fig. 9. The

slightly positive correlation of the collection efficiency with the electrical mobility size was found. As shown in Fig. 9, the

collection efficiency at the sampling flow rate of 0.3 L/min achieved ~100% as the increase of particle size. In the case of 0.6

L/min. the collection slightly increased with the increase of electrical mobility particle size. The experiment results are similar
to Beresnev et al. (2019). It is known that soot particles are agglomerates of primary particles. Their thermal precipitation

velocity cannot be estimated by Eq. (3-4) because the equation assumes particles are solid and in spherical shape. For soot

particles, its density and thermal conductivity are very different from the bulk material, and its shapes are not spherical. We

suspected the effect of particle shape may play an important role in the thermal precipitation of soot particles. It is because the

collection efficiency of 23nm soot particles at the sampling flow rate of 0.6L/min was 46.4% which approximately equals to
that of NaCl particles (45.9%) of the same electrical mobility size. At a small mobility size, soot agglomerates are structured

by a few of primary particles.

## 5. Conclusion

A new type of aerosol electrometer, i.e., thermal precipitation aerosol electrometer (TPAE), has been developed. Its overall

performance has been experimentally calibrated and compared it with that offered by the reference (TSI aerosol electrometer).
The TPAE integrates the thermal precipitation chamber with a micro-current measurement circuit. The precipitation chamber

is in the disk-to-disk configuration and its temperature gradient was established by heating the top disk and cooling of the

bottom disk. Air was selected as the cooling flow in TPAE instead of liquids used in previous works. A current probe (i.e.,

solid cupper pin) of the micro- current measurement circuit was attached to the cold disk (converting it to an electrode) which

is enclosed by the top disk and cooling flow chamber to minimize the potential interference from the ambient electromagnetic
waves.

For the performance calibration, the zero-point of the prototype was first measured from the warm-up to the stable operation.

The measurement of TPAE response time was also conducted and compared it with that of the reference. It is found that the

different between both electrometers was within one second. The linear correlation between the readouts of both aerosol

electrometers was further confirmed.

The collection efficiency of TPAE was experimentally investigated. It is found the effects of temperature gradient, sampling

flow rate and particle size on the particle collection efficiency are consistent with those obtained from the previous model and

experimental data. It provides solid evidence for the successful development of this new aerosol electrometer. In addition to NaCl particles, soot particles were also used in the collection efficiency measurements. It is found that the collection efficiency of soot particles was slightly increased as the increase of mobility particle sizes at the given setting of sampling flow rate and

temperature gradient, whose trend is different from that of NaCl particles. It is possibly because soot particles in large mobility sizes are agglomerates of primary particles instead of solid and spherical particles (assumed by the models). Soot agglomerates have different density and thermal conductivity compared with those of bulk materials.

**Data availability.**

Requests for all data in this study and any questions regarding the data can be directed to Shipeng Kang

(spkang@mail.ustc.edu.cn).

**Competing interests.**

The authors declare that they have no conflict of interest.

**Special issue statement.**

This article is part of the special issue "In-depth study of the atmospheric chemistry over the Tibetan Plateau: measurement,

processing, and the impacts on climate and air quality (ACP/AMT inter-journal SI)". It is not associated with a conference.

**Author contributions.**

SK: Writing – original draft, Visualization, Data curation and analysis; TY: Sample collection, Project administration; YY: Resources, Data validation; JW: Experiment; HG: Writing – review; JL: Conceptualization, Supervision; DC: Writing – review & editing, Experiment, Formal analysis

**Acknowledgements.**

This research was supported by the National Natural Science Foundation of China (42005108), the Science and Technological Fund of Anhui Province (2008085MD116), Major Subject of Science and Technology of Anhui Province (202003a07020005), Strategic Priority Research Program of the Chinese Academy of Sciences (XDA23010200), the National Engineering Laboratory for Mobile Source Emission Control Technology (NELMS2020A09) and the HFIPS Director's Fund (Nos.



BJPY2021A04).

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
