# Peer review of "Design and Evaluation of a Thermal Precipitation Aerosol Electrometer (TPAE)"

_Atmospheric Measurement Techniques, 2023_

## Author Comment (AC1)

General comments:

1. What is the motivation for designing this new electrometer? Compared with the TSI electrometer, why do we need this TPAE?

Reply: Thanks for the question. This development was motivated by the fact that TSI electrometer can only measure the aerosol current. The recovery of collected particles for future offline characterization such as the particle morphology by SEM and chemical analysis is not possible. Although ELPI can be used to collect particles, the collection by the inertial impaction may alter the shape of particle agglomerates, e.g., soot particles. By combining thermal precipitation with aerosol current detection, the device can measure the aerosol current while collecting particles by soft landing. Moreover, the requirement of changing HEPA filters used in existing AEs is not needed (a welcome for instruments to be used the field study).

The above advantages have been stated in Line 29~31 of revised manuscript.

The motivation had been added in Line 71~73 of revised manuscript: "*The collection of charged particles by the filtration, inertial impaction ... is not favored for the off-line SEM analysis if required*".

2. Experimental design issues

(1) The singly charged assumption limits the application of this electrometer. Therefore, the author should consider adding a CPC or another SMPS in the setup to measure the charge distribution for the testing particles. Then implement that into the calculation in equations 1-3. Or at least provide the uncertainty caused by the single charge assumption.

Reply: Thanks for your suggestion. Because our test particles were obtained by the DMA classification, it is the reason why we assumed particles are singly charged. For the general application of the device, a particle charger must be used in front of the developed device. The charge status of particles is thus dependent on the charging performance of selected chargers. Accordingly, Eq. 2 has been modified to include "$x$ is the average charge of particles, which shall be obtained for the charger calibration (Line 123). The accuracy of $x$ did affect the accuracy in the conversion from the aerosol current to the particle number concentration. However, the performance of the TPAE would not be affected.

(2) The size effect on the performance of the TPAE covered 25-200 nm. Are those results under the singly charged assumption? If so, will the charge distribution affect the result?

Reply: Thanks for the question. The result shown in the manuscript were under the singly charged particle assumption. In our experiments, the fraction of multiple-charged particles was minimized by selecting test particles from the right-hand side of the peak size in the size distribution of particles generated from aerosol generators. In addition, the ratio of multiple charged particles to singly charged ones is low (lower than 2.1%, $d_p \leq 40nm$) according to Wiedensohler Formula, as shown

in Figure R1. The combination of the above factors will significantly reduce the fraction of multiple-charged particles in test particles.

**Table 1-1**
Distribution of Charges on Aerosol Particles According to the Wiedensohler Formula

| | Percent of Particle Carrying Np Elementary Charge Units | | | | | | | | | | | | |
|---|---|---|---|---|---|---|---|---|---|---|---|---|---|
| Dp(µm) | Np=−6 | −5 | −4 | −3 | −2 | −1 | 0 | +1 | +2 | +3 | +4 | +5 | +6 |
| 0.01 | | | | | | 5.14 | 90.75 | 4.11 | | | | | |
| 0.02 | | | | | 0.02 | 10.96 | 80.57 | 8.64 | 0.01 | | | | |
| 0.04 | | | | | 0.54 | 19.50 | 64.79 | 14.86 | 0.31 | | | | |
| 0.06 | | | | 0.02 | 1.92 | 24.32 | 54.13 | 18.51 | 1.09 | 0.01 | | | |
| 0.08 | | | | 0.11 | 3.73 | 26.81 | 46.75 | 20.46 | 2.10 | 0.05 | | | |
| 0.10 | | | | 0.37 | 5.63 | 27.31 | 42.28 | 20.91 | 3.30 | 0.17 | | | |
| 0.20 | | 0.05 | 0.53 | 3.40 | 12.38 | 25.49 | 29.66 | 19.51 | 7.26 | 1.53 | 0.18 | 0.01 | |
| 0.40 | 0.27 | 1.14 | 3.60 | 8.54 | 15.24 | 20.46 | 20.65 | 15.66 | 8.93 | 3.83 | 1.24 | 0.03 | 0.05 |
| 0.60 | 1.21 | 3.00 | 6.19 | 10.53 | 14.82 | 17.25 | 16.60 | 13.20 | 8.69 | 4.73 | 2.13 | 0.79 | 0.24 |
| 0.80 | 2.42 | 4.64 | 7.71 | 11.12 | 13.90 | 15.06 | 14.15 | 11.53 | 8.15 | 4.99 | 2.65 | 1.22 | 0.49 |
| 1.00 | 3.56 | 5.84 | 8.53 | 11.13 | 12.96 | 13.45 | 12.46 | 10.30 | 7.59 | 5.00 | 2.93 | 1.54 | 0.92 |

Figure R1. Charge fraction given in TSI3088 (X-ray Neutralizer) manual

(3) This study didn't characterize the real concentration range of the electrometer. At least, it should give the lowest limits for reliable concentration detection for different sizes of particles.

Reply: Thanks for your suggestion. The lowest limits for reliable concentration detection for particles are decided by $C = \frac{\Delta N_p}{q_s t_s} = \frac{\int \Delta I_p dt}{exn q_s t_s}$, where $C$ is the number concentration of sampled particles, $t_s$ is sample time and $q_s$ is sample flowrate. The concentration range of the device does not depend on the particle size directly.

By the calculation, the number concentration range of TPAE is $2,500 \#/cm^3 \sim 6.25 \times 10^7 \#/cm^3$ at $q_s = 0.3L/min$ and $x = 1$. The following is the calculation: TPAE collects 98.9% of particles at the sampling flow rate of 0.3 lpm according to Figure 5(a). Thus, with $q_s = 0.3L/min$, $\eta = 98.9\%$, $x = 1$, $t_s = 1s$, we obtained $C \approx (1250cm^{-3} \cdot fA^{-1})\Delta I_p$. The lower limit of aerosol concentration is based on the lower limit of current detection, $\pm 2fA$, resulting in the number concentration of $2,500 \#/cm^3$. The upper limit of particle concentration was calculated based on the upper limit of current detectable by the used micro-circuit, which is $\Delta I_p = 5 \times 10^4 fA$.

The upper limit range of TPAE is difficult to be experimentally determined because the concentration of test particles (after a DMA) is very low compared to the required concentration ($6.25 \times 10^7 \#/cm^3$).

The above information is included in the abstract (Line 22-23) of the revised manuscript.

Specific comments:

Abstract: What are the "other physical parameters"? It might be good to mention them here to emphasize the need for TAPE development.

Reply: Thanks for the question. Instead of "other physical parameters", we have revised the paragraph as "Note that the charger-AE assembly……." (Line 51-54).

Line 22: Is this range based on theoretical estimation or experimental confirmation? It seemed that this was only a theoretical range. If so, please provide a practical range. In addition, it is even desirable to convert it to the aerosol concentration range. Please also specify the corresponding size range.

Reply: Thanks for the question. Please see our detailed reply in Q#3. The range is based on theoretical estimation. The calculated number concentration range has been added in Line 22-23 of revised manuscript.

Line 24: The evaluation used one TSI aerosol electrometer to determine another aerosol electrometer – TAPE? Why not use CPC? How does the author determine the charging state for the testing particles?

Reply: Thanks for the question. The reason for not using CPCs for the comparison is because electrometers are used to calibrate the performance of CPCs (ISO 27891:2015). The number concentration of particles ($\#/cm^3$) must be traced to the current (fA) eventually.

Test particles are assumed to be singly charged because of the use of a DMA for particle classification. The DMA classification of particles cannot completely avoid the inclusion of multiple-charged particles in DMA-classified particles. To minimize the fraction of multiple-charged particles, we classified test particles from the right-hand side of the peak size of particles generated from the aerosol generators. The details can be found in our reply of Q#2.

Line 27: "The effect of particle size on the above efficiency was minor for sodium chloride particles." This is not true in general. It should depend on the size range of the testing particles.

Reply: Thanks for the question. The conclusion arrived according to our experimental results, Figure 8, in addition to previous studies on the thermal precipitation of sub-micro-meter particles (such as http://dx.doi.org/10.1016/j.jaerosci.2012.04.004: Performance study of a disk-to-disk thermal precipitator). Please note that "minor" does not mean "no".

Accordingly, the description has been revised as "The effect of particle size on the above efficiency was minor for sodium chloride particles in the sizes of 23-200 nm" (Line 27-28).

Line 55 and 58, please spell out DiSC and TEOM.

Reply: Thank you for the suggestion. DiSC (Diffusion Size Classifier), TEOM (Tapered Element Oscillating Microbalance). The above has been added to Line 59 and 63.

Line 73: How does the author characterize the aerosol particles as "small" or "large"? Please specify the size range.

Reply: Thank you. "Small" and "large" are adjectives for the particle sizes. There is no scientific

consensus in the aerosol community on how "small" is considered as small and how "large" is considered as "large". By electrical classification and detection, "small" particles are for ones of $d_p < 800nm$ while "large" is for particles of $d_p \geq 800nm$. The above is based on the upper size limit for TSI DMA classification.

Accordingly, the above statement has been revised as "the collection by the inertial impaction favors for inertial particles, and the electrical collection favors for diffusive particles' (Line 79)

Introduction: It is not clear what the advantages are of developing a TPAE. Will it extend the lower detection efficiency for small-size particles? Or will it cover a wide range of aerosols?

Reply: Thanks for the question. Please see the detailed reply in Q#1

Fig. 1: Where are the thermistors? And pre-amplifier? Please add them to the figure. Will the location affect the temperature control? If so, please explain the effects. How do you maintain the thermal gradient under different environmental conditions? Are there any feedbacks to control the cooling flow or heating plate? Will the device be used in an outdoor environment?

Reply: Thanks for the question. The thermistors were black arrows in Figure 1 and the captions in Figure 1 had pointed it out. The "amplifier" in Figure 1 has been revised as "pre-amplifier".

The uniformity of plate temperature was examined (using a thermocouple probe) prior to our testing. The variation of plate temperature was within three degrees (°C). In our lab, the room temperature was relatively stable. As the heating and cooling power of the device were fixed, the temperature gradient in the precipitation zone eventually reached stable. Therefore, the temperature feedback control was not conducted. For the outdoor applications, feedback control on the plate temperature will be included.

Fig 7. Does the fitting line indicate that the efficiency is higher than 120% at 0.2? Is it realistic? Why do we have different fitting parameters for different size aerosol particles? Such as 30 for 70 nm and 26.3 for 200 nm? It is crucial to investigate the size effect on the performance of this electrometer.

Reply: Thanks for your correction. The maximal collection efficiency is 100%. Accordling, the figure has been corrected.

The thermophoresis force is calculated by $\vec{F}_{th} = \frac{3\pi\mu^2 d_p H \overrightarrow{\Delta T}}{\rho_g T}$, and drag force is calculated by

$\vec{F}_{drag} = -\frac{3\pi\mu d_p \vec{V}}{C_c}$, where $H = \frac{2C_s(\frac{k_g}{k_p}+C_t K_n)}{(1+3C_m K_n)(1+\frac{2k_g}{k_p}+2C_s K_n)}$. $\overrightarrow{\Delta T}$ is temperature gradient, $T$ is the

absolute temperature of particle, $\rho_g$ is the density of air and $H$ is the thermophoretic coefficient.

Let $\vec{F}_{th} + \vec{F}_{drag} = 0$, we could get $\vec{V}_{th} = \frac{\mu \overrightarrow{\Delta T} H C_c}{\rho_g T}$, according to which we know that $\vec{V}_{th}$ is not

directly related to $d_p$, while $H$ and $C_c$ are related to $d_p$. Thus, the effect of $d_p$ on the thermal

precipitation does exist but is minor. Consequently, the fitting result should not be the same. The size effect on the thermal precipitation was shown in Figs. 8 and 9.

Fig 8 and Fig 9, why several typos in the manuscript: TDAE? Should it be TPAE?

Reply: Thanks. It is corrected.

---

## Author Comment (AC2)

The manuscript describes the evaluation of a novel thermal precipitation based aerosol electrometer (TPAE) that simultaneously collects particles for subsequent analyses while measuring the current stemming from the deposition of charged particles. If the charge level of the particles is known, e.g. when used downstream of a DMA, the number concentration of particles can be determined. I find the device very interesting and the evaluation has thoroughly been carried out. However, I have a several major and a few minor concerns that I believe the authors should address in a revision:

Major concerns:

It does not become very clear, what is main field of application for the device. Is it mainly to be used downstream of a DMA? This would guarantee that particles are mostly singly charged and that the number concentration can be derived from the current measurement. But then is the main purpose to collect particles for subsequent analyses or to measure particle number concentrations and/or size distributions?

Reply: Thanks for the question. Like existing aerosol electrometers (equipped with HEPA filters), TPAE requires working with charged particles. An aerosol charger is required to charge particles prior to TPAE. The total number concentration of particles can be measured if the average charge status of particles after the charger is known. The assembly of charger-electrometer has also been applied for the particle surface and mass concentration measurement. In the above cases, empirical calibration of assembled devices using typical particles is required and the calibration curves may be varied when measuring particles in the chemical composition different from that of calibration particles. When an electrical mobility classifier/analyzer (DMC/DMA) is included between the charger and aerosol electrometer, the size distribution of particles can be obtained after the correct of multiple charges. The new feature of TPAE is the soft collection of sub micrometer particles, enabling the offline characterization of particle morphology and chemical composition. The above feature is desired when measuring particle agglomerates. The other advantage of the TPAE is without the requirement of periodically changing filters.

When size distributions are measured, how much does the "smearing effect" of the particle deposition inside the device due to the residence time of the particles in the gap space between the two plates before being deposited affect the accuracy? This effect is also likely the reason for the differences observed in the dynamic responses in Figure 4.

Reply: Thanks for the question. Working with a DMA (differential mobility analyzer), particle smearing occurs during the transportation of classified particles from the DMA exit to the collection of particles in the TPAE. The transportation time in TPAE can be calculated in the following: Taking $Q$ as the flowrate of sample flow, $h$ as the distance between two plates, $x$ as the distance between particle position and center axis of disk, $t$ is the residence time of particle in the gap space, it could be calculated that

$$\frac{Q}{2\pi x h} \cdot \mathrm{d}t = \mathrm{d}x$$

$$dt = \frac{2\pi x h}{Q} \cdot dx$$

$$t = \frac{\pi h}{Q} x^2 \bigg|_{x=60mm} - \frac{\pi h}{Q} x^2 \bigg|_{x=0mm} = \frac{3.14 \times 0.5mm \times 60^2 mm^2}{0.3 \times \dfrac{10^6 mm^3}{min}} = 0.019min = 1.13s$$

When $Q$ is 0.3L/min, the maximal particle residence time in TPAE is 1.13s.

Figure 4 shows the overall response time of TPAE including the particle residence time in TPAE and the response time of micro-current measurement circuit.

Note that the particle smearing effect can be minimized when the DMA is operated at the voltage stepping mode instead of voltage scanning mode.

It is not mentioned what size range the instrument is designed for. I assume, only for particles smaller than approximately 300 nm, for which nearly size-independent thermophoretic deposition can be expected. It is also not discussed that the assumption of size-independence only holds in the free molecular and near free-molecular regime, where the thermophoretic force (see L. Waldmann and K. H. Schmitt, *Aerosol Science*, edited by C. N. Davies, Academic, New York, 1966) and the opposing drag force share the same size dependence.

Reply: Thanks for the comment. The TPAE is designed for particles in sizes smaller than 200 nm. Accordingly, the size range description has been included in the revised manuscript (Line 17*)*.

When the TPAE is used downstream of a DMA, the size-independence of the deposition only holds for a fixed DMA-voltage. When the voltage is ramped during size distribution measurements, the concentration and thus the representativeness of the collected sample is dependent on the particle size due to the size dependent particle charging probability.

Reply: Thanks for the comment. Indeed, for the offline characterization of particles, the DMA voltage requires to be fixed. When the DMA+TPAE is used for the size distribution measurement, the smearing of classified particles can be prevented by operating the DMA at the voltage stepping mode instead at the voltage ramping mode.

Why was a radial preferred over a rectangular design? While the deposition efficiency is nearly independent of particle size (with the limitations mentioned above), the radial design introduces a spatial dependence on the particle deposition, due to the decreasing flow velocity from the center to outside. This could be overcome by a rectangular design (see e.g. Azong Wara et al., *J. Nanopart. Res.* **11**: 1611-1624, 2009)

Reply: Thanks for the comment. The thermal precipitation can be done either in the parallel-plate, parallel-disk and coaxial tube configurations. The selection of the radial

configuration for TPAE was because it will be a part of the particle sizer under the development and the radial design best fits the limited space requirement. The uniformity of particle deposition is not our concern.

The suggestion is, however, excellent. We will investigate the rectangular configuration of TPAE in the near future.

Why was air-cooling preferred over the more commonly used liquid cooling?

Reply: Thanks for the question. The primary reasons for using air cooling in TPAE (instead of liquid cooling) is to suppress the noise of current signal, and (2) to reduce the liquid handling for the TPAE operation (convenient for field applications). Liquid cooling is used in published thermal precipitators because of the high specific heat of liquids (carrying more heat). However, the current input needle of TPAE is directly connected to the TPAE cold plate. If a liquid is used for colling, the contact of cold plate and liquid (with the liquid conductivity much higher than air) significantly increases the noise of current signal. The other reason for air colling is to remove the liquid handling for the TPAE operation (a welcome for field applications).

With such rather high temperature gradients, did you check if buoyancy may affect the deposition efficiency?

Reply: Thanks for the question. The buoyancy was not concerned because of the tiny spacing between two disks (i.e., only 0.5 mm). With the tiny spacing, the temperature difference between two disks is small to achieve the high temperature gradient.

Section 4.1.2 provides only observations. I would expect to see at least some quantitative analysis, such as the $t_{10\text{-}90}$ and $t_{90\text{-}10}$ time constants of both the TSI electrometer and the TPAE.

Reply: Thanks for your suggestion. For sample flowrate of 0.3L/min and 0.6L/min, $t_{10\text{-}90}$ and $t_{90\text{-}10}$ are calculated for TPAE and TSI3068B.

For 0.3L/min, TPAE: $t_{10\text{-}90}$=36.42s-32.46s=3.96s; TSI: $t_{10\text{-}90}$=36.23s-32.4s=3.83s.

For 0.3L/min, TPAE: $t_{90\text{-}10}$=134.09s-127.5s=6.59s; TSI: $t_{90\text{-}10}$=134.01s-127.38s=6.63s.

For 0.6L/min, TPAE: $t_{10\text{-}90}$=20.96s-19.13s=1.83s; TSI: $t_{10\text{-}90}$=21.53s-19.45s=2.08s.

For 0.6L/min, TPAE: $t_{90\text{-}10}$=77.21s-74.56s=2.65s; TSI: $t_{90\text{-}10}$=78.58s-75.06s=3.52s

For 0.3L/min, the response time of TPAE almost equaled to TSI3068B while for 0.6L/min, TPAE responded faster.

The above data has been added in Line 256~260 of revised manuscript.

While the manuscript in principle well written, it would benefit from a language check by a native speaker.

Reply: Thanks for your suggestion. The revised manuscript has been checked by a native speaker with a PhD degree in engineering.

Minor comments:

Line 51: what is meant with "early generation" of TSI model 3068B? To the best of my knowledge, this model is still commercially available.

Reply: Thanks for the comment. The description has been deleted.

Line 52: it should read Faraday cup, not Faraday cage

Reply: Thanks for the comment. It has been corrected.

Line 55: This paper refers to the mini DiSC, not DiSC

Reply: Thanks for the comment. It has been corrected.

Page 3, first paragraph: The Nanomater Aerosol Sampler (TSI model 3089, Dixkens et al., *Aerosol Sci. Technol.* **30**: 438-453, 1999) is missing in the discussion. This has been developed to electrostatically sample monodisperse particles downstream of a DMA.

Reply: The missing reference has been added in Line 71-72 of the revised mansucript.

Line 77: particles move in counter-direction of a thermal gradient, not in the direction.

Reply: Thanks. It has been corrected.

Line 100: it appears that height and diameter are reversed here; it should thus read …120 and 0.5 mm in diameter and height, respectively. I would also suggest the term "gap distance" rather than height.

Reply: Thanks for the suggestion. It has been revised.

Line 220-225: When I calculate the temperature gradient based on the provided values, I get 265 K/cm, not 254 K/cm. Either the temperature values or the value for the gradient seems to be incorrect.

Reply: Thanks for pointing it out. Accordingly, it has been corrected throughout the revised mansucript.